# Advice on Regulating Body Mass in Wrestling from the Most Cited Combat Sport Literature—A Systematic Review

**DOI:** 10.3390/jfmk9040264

**Published:** 2024-12-09

**Authors:** Hrvoje Karninčić, Nenad Žugaj, Krešo Škugor

**Affiliations:** 1Faculty of Kinsiology, University of Split, 21000 Split, Croatia; kresoskugor95@gmail.com; 2Faculty of Kinsiology, University of Zagreb, 10000 Zagreb, Croatia; nenadzugaj@yahoo.com

**Keywords:** combat sports, weight reduction, health

## Abstract

**Background**: Since studies on the subject of weight reduction in wrestlers vary in opinions regarding health, performance, and legal regulations, a review of the most cited literature in other combat sports was carried out. **Methods**: By searching the WOS and Scopus scientific databases, the top 60 cited papers were identified, and ultimately, 13 articles that met the inclusion criteria for this review were selected. **Findings and conclusions**: Apart from the advantages gained in strength and mass through weight reduction, a mental advantage is also obtained. The period between weigh-ins and the actual bouts is considered crucial for the preservation of performance. When the rules allow for a longer recovery period, this enables aggressive, harmful, or prohibited weight reduction methods to be employed by some wrestlers. A chronic reduction in body weight is regarded as a long-term health issue but may be ignored by wrestlers. There has been no progress in the attempt to regulate the problem of rapid weight loss (RWL) within wrestling in the past thirty-five years. If any change is to be brought about, the engagement of all federations, clubs, WADA, and all individuals involved in this issue (athletes, coaches, parents, doctors, etc.) is deemed necessary.

## 1. Introduction

The regulation of body weight is an important part of wrestling. Weight categories were introduced in wrestling to ensure fair play or bouts between opponents that are evenly matched in physical size. Having multiple weight categories has been a practice since the very beginning of modern wrestling. It was only at the first Olympic Games (Athens, 1896) that there was one weight category. At the World Championship in Vienna, Austria, in 1904, wrestlers were divided into two weight categories (middleweight and heavyweight), and at the Olympic Games in St. Louis, USA, there were seven weight categories [1]. Since the introduction of weight categories, the phenomenon of bodyweight management has emerged. Wrestlers strive to compete in the lowest weight category possible to gain an advantage in strength. It did not take long for this phenomenon to be recognized as a problem, and it can be found in the scientific literature as early as the 1930s [2]. The effects of diet and dehydration on a wrestler’s body have been investigated [3]. Today, almost 100 years later, this issue still remains relevant in the literature. The number and arrangement of weight categories have changed, but harmful weight management practices persist. In body mass regulation in wrestling, special emphasis is placed on rapid weight loss (RWL) and rapid weight gain (RWG) due to their aggressive nature and the manipulation of a significant amount of body mass in a very short time.

Although this problem is most pronounced in combat sports, it also exists in some other sports. This is why a plethora of scientific papers have been written on this topic (at the time of writing this paper, the Web of Science scientific database (Core Collection) produced 2938 results in a search using the terms “weight reduction” and “sport”). Today, the challenge is not accessing research on the topic of body weight reduction in wrestling but rather navigating through the data and papers to select relevant ones and draw conclusions that can be helpful in practice (for wrestlers, coaches, and other professionals). In a previous review, a search for scientific papers exclusively related to body weight reduction among wrestlers (Scopus and WoS databases) yielded 70 papers [4]. These papers focusing exclusively on wrestlers were categorized into five main themes: weight loss and health, weight loss and performance, weight loss habits, weight loss and legal regulations, and weight loss psychology [4]. Papers addressing the health aspect were found to be divided on their opinions concerning the harmfulness of RWL to wrestlers’ health. There are studies that stated that RWL does not negatively impact growth and development; hormonal status adapts rapidly to RWL [5]; and RWL does not affect eating disorders [6,7], blood markers [8], or metabolism [9]. It is interesting that almost the same number of studies point out the health risks for wrestlers. These studies reported that RWL leads to a state of severe dehydration [10,11,12], negatively affects metabolism [13,14], and has a negative impact on wrestler injuries [15,16], on the hormonal status of female wrestlers [17], on body composition [18,19], on stress markers [20], and on eating disorders [21]. It is very difficult to draw a conclusion about how harmful RWL is to health when there are arguments on both sides. A similar situation arises in the relationship between RWL and wrestling performance. There are many studies in which their authors did not observe a decline in performance caused by RWL [22,23,24,25,26,27] or a rapid recovery of performance after RWL [24,28,29,30]. There is an equal number of scientific studies in which the authors observed a decline in performance, results, and even techniques caused by RWL [31,32,33,34,35,36,37,38]. The third topic that has interested researchers is weight reduction habits in wrestling. Unlike in the other topics, authors unanimously agree that wrestlers extensively use unhealthy methods for weight reduction, losing significant amounts of body weight and starting the reduction process prematurely. They also concur that there are not enough studies to recommend healthy weight reduction methods [11,39]. Based on the number of studies, the fourth topic is “weight loss and legal regulation in wrestling”. In the United States, legal regulation specifying the minimum weight category in which a wrestler can compete has existed since 1989. Some of the authors believe that this approach yields good results, that the legal framework is sound, and that it can be further improved, advocating for its implementation everywhere. However, a few authors still believe that wrestlers find ways to bypass this regulation, meaning that the legal framework has not yielded the desired results [40,41,42]. The last topic with the fewest studies is “weight loss psychology”. Only a few studies have investigated this subject. During RWL, wrestlers experience mild depression and increased confusion, but with experience, they cope better with RWL [43].

Studying the phenomenon of weight regulation exclusively within the wrestling population has not helped us draw conclusions due to divided opinions and mixed scientific findings. Reviewing only scientific studies with wrestler samples has a significant limitation. Studies on related combat sports are not considered, but they are also important in the scientific literature. The aim of this paper is to review highly cited studies dealing with weight regulation in combat sports. These studies should help in drawing conclusions and making recommendations for wrestlers who manipulate their body weight. This paper summarizes a topic that has been written about for nearly 100 years and investigated in thousands of studies.

## 2. Materials and Methods

To be included in this review, intervention studies involving animals or humans and other studies that required ethical approval had to list the authority that provided approval and the corresponding ethical approval code. We searched the scientific databases Web of Science (WOS) and Scopus, which are considered the most relevant databases in the field of sports science. In addition to the fact that they are relevant databases, they were also selected because they offer the possibility of classifying works according to the number of citations. The primary keywords used in all searches included weight control, weight management, weight cycling, weight loss, rapid weight loss, and rapid weight gain. The secondary keyword used in all searches was combat sport. Procedure: Example WOS: A combination of two keywords (weight control martial arts) was entered; the search returned 154 results; the results were sorted by citations (drop-down menu in Wos), the most cited first. The five studies with the most citations were included in the further process. Of the five papers obtained, papers 2 and 3 met the criteria of this study. We have 6 combinations of two keywords, from each of which we select the 5 studies with the most citations. In each search, the top 5 most cited papers were selected, resulting in a total of 60 papers from two databases (Wos, Scopus). The exclusion criteria were as follows: duplicate papers; papers not about weight regulation; papers not about combat sport. The inclusion criteria were as follows: full paper format, English language. The database searches were carried out by all authors. The inclusion and exclusion criteria were applied to 60 papers collected from the databases. A table with all 60 papers and an explanation of the criteria used to exclude some papers from this review can be found in the Appendix A. After the exclusion criteria, 13 papers remained, all of them in full-text format and all in English, so the final number of papers was 13. The final papers included in this study were grouped into 5 categories (weight loss habits, weight regulation and health, weight regulation and performance, legal regulation, and psychology of weight loss). Each author analyzed one or two categories, followed by a review by an author who did not analyze that category. The final conclusions for each category were reached by consensus of all authors. All stages of the literature selection were recorded using a PRISMA flow diagram. The protocol of this review is registered in the INPLASY register. (registration number INPLASY202470007; DOI 10.37766/inplasy2024.7.0007). The searches included the most cited papers on body mass regulation in combat sports, and other methods of quality control were not applicable. The PRISMA checklist and the table of applied criteria for all 60 papers can be found in the Appendix A.

In Figure 1, we can see that out of the total number (60 papers), only 13 papers met the inclusion criteria for this review study.

## 3. Results

As shown in Table 1, Table 2 and Table 3, we can see that Franchini, Artioli, and Raele are the top highly cited authors, and all of their papers were published within ten years from 2009 to 2019. Seven studies were conducted on samples from various combat sports, whereas three studies were conducted on samples from both judo and MMA (mixed martial arts). These authors primarily wrote about weight reduction habits, followed by their impact on performance. Health psychology and legal regulations are represented by two papers each.

## 4. Discussion

Thus far, the reasons why athletes resort to RWL and RWG include an advantage in strength and an advantage in weight. However, two more reasons can be found in the most cited literature: a mental advantage (which will be further explained in a psychology review) and issues with moving up to a higher weight category. In younger age categories, category changes are a normal occurrence that are driven by growth and development. However, seniors strive to wrestle in the same category for as long as possible. Franchini states that few athletes have successfully changed weight categories, and new opponents also require a new tactical battle plan [44]. Regarding methods and habits of weight reduction, all authors agree that athletes often reduce excessive amounts of weight in a short time, using harmful and sometimes prohibited methods, even among younger athletes [44,45,46,48,51].

**Health issues:** The detrimental impact of RWL on wrestlers’ health is unquestionable, but at the same time, rapid recovery and restoration of disrupted physiological parameters to normal levels have been observed [5,8,57]. Malnutrition during the competitive season can disrupt the function of growth-related hormones, but after the season, parameters quickly return to normal values [5]. In high school wrestlers (13 years old) who lost body mass rapidly and quickly regained it, all biomarkers (sodium, potassium, chloride, hematocrit, and hemoglobin) were in the normal range at the pre-match weigh-in [8]. The magnitude and cyclic method of weight loss used by wrestlers does not appear to prevent myocardial hypertrophy [57]. The most cited papers do not provide new insights regarding the effects of RWL on health, except for general effects. Severe hypohydration during RWL leads to a reduction in blood plasma volume (which affects cardiovascular efficiency); dehydration impairs the body’s thermoregulatory system (which is important when exercising in such a state); RWL promotes hormonal imbalance and suppresses immune function [51]. Franchini (2012) states that athletes who begin losing weight at an early age are exposed to a higher risk of weight loss-related problems [44]. Artioli (2010) states that athletes who start weight reduction early tend to resort to more extreme reductions [45]. Artioli further reports that Kiningham and Gorenflo (2001) found that an earlier entry into weight reduction leads to athlete adaptation and performance preservation [58]. This should imply that an early initiation of RWL can lead to the body adapting to such conditions, and negative consequences will not impact performance. However, an early start in RWL results in larger weight reductions and the use of more aggressive methods, thereby causing greater long-term harm to athletes’ health. The idea that wrestlers should gradually enter weight reduction at a younger age to build adaptive mechanisms is questionable, considering the health risks involved.

**Performance issues:** The answer to the question of whether performance is affected by RWL is complex. Many authors have recorded a decline in performance due to RWL, but an equal number of authors have not observed such a decline. The most cited papers on this topic suggest that an early entry into weight cycling leads to adaptive changes and that athletes undergoing this process experience less performance loss [47,58]. However, Mendes (2013) contradicts this notion [54], finding that athletes with extensive experience in weight reduction did not recover their performance better than the control group. Better recovery was associated with the time between weighing and competition, as well as with food and beverage intake [54]. There are several hypotheses about adaptive changes in chronic weight cyclers (chronic weight cyclers have a slowed loss of glycogen during food restriction, an increased rate of glycogen resynthesis when resuming food intake after dieting, and an increased ability of the body to function in a dehydrated state). Mendes tested these hypotheses and found no adaptive changes in his experiment [54]. In contrast to Mendes (2013), Artioli (2010) finds a difference between experienced and inexperienced judokas. Experienced judokas can lose 5% of their body weight without this affecting their performance in judo [47]. The 4 h break between weigh-in and competition is sufficient to compensate for food and fluids. The special adaptation of experienced fighters manifests itself in the rapid storage of glucose in the body. The group that lost weight during the break (4 h) consumed large amounts of food and drink. During the same break, there was a significant drop in blood glucose levels. This illogicality can only be explained by the adaptation of the body of fighters with a lot of experience in losing weight. Experienced fighters manage to store large amounts of glucose from food in the form of glycogen within a very short time and are ready to fight again after an exhaustive reduction in body mass. This hypothesis was not confirmed by a biopsy, insights into the state of glycogen in the muscle itself, and the control of enzyme activity [47]. Mendes (2013) cites a number of studies where there is a serious decline in performance when there is no recovery time (time between weigh-ins and first fight). Performance decline is inevitable in RWL; time plays a key role in recovery, and 4 h is sufficient for experienced fighters [47]. The question is whether the use of RWL would decrease if recovery time were completely abolished. We can conclude that those who start weight reduction earlier may not develop adaptive mechanisms but rather better strategies for managing weight reduction, recovery, and psychological skills.

**Psychological issues:** Studies focusing on the psychology of RWL among wrestlers suggest milder issues: increased confusion, tension, anger, depression, fatigue, and reduced concentration and short-term memory [59,60]. However, these feelings change with food and beverage intake, and as wrestlers become older, they experience fewer difficulties [43]. With older age, wrestlers become able to channel negative emotions (anger and tension) into positive stimuli and improve their performance [61]. Costarelli (2009) reports that athletes who resort to RWL have higher levels of emotional intelligence and a better self-image; they do not develop eating disorders, as these disorders are linked to anxiety [55]. A particularly interesting study is a study conducted by Pettersson (2013), who associates RWL with a mental advantage over opponents. Fighters who undergo RWL develop a sense of group cohesion (sport identity) and enhance their focus, discipline, and control (diversion of negative thoughts and emotions), while those who do not undergo weight reduction feel disadvantaged (signal power) [49].

**Weight loss habits:** Regarding habits associated with RWL, all studies on wrestlers, including the most cited papers, unanimously agree that weight reduction methods range from harmful to prohibited [21,44,45,46,48,51,58]. The most commonly used unhealthy methods are as follows: major restrictions in eating and drinking, skipping one or two meals, increased exercise, sauna or plastic clothing, exercising in a hot environment, diuretics or laxatives, diet pills, and spitting or vomiting [45]. The problem is further complicated when developing children use these methods. Premature reductions in body mass occur in all martial arts but are most common in the striking martial arts [46]. Concerning prohibited methods, more recent findings reported in the most cited articles show that the use of diuretics is the most common cause of doping scandals in combat sports [44], and the banned use of intravenous infusion of saline solution (to rehydrate the body and increase weight) is associated with the time between weighing and competition [47]. This raises a very important dilemma of whether to allow enough time between weighing and competition to enable wrestlers who have reduced significant amounts of body weight to recover. In MMA, there is typically more time available between weigh-ins and the ensuing fights. A previous study by Matthews reported that MMA fighters lost an average of 5.6 kg over 7 days, but within 32 h (the time between weighing and their fight), they gained an average of 7.4 kg [52]. It is clear that a longer period between weigh-ins and the ensuing fights leads to greater unfairness in combat sports.

**Legal regulation:** Several of the most cited papers state that RWL or RWG is cheating and has no place in sports. Rapid weight loss (RWL) is losing more than 5% of body weight in less than 7 days. Most fighters use this method of regulating body mass. In general, 60% of fighters (of various combat sports) use RWL [46]. This percentage is even higher at higher levels of competition. Steen and Brownell (1990) state that 98% of the US Olympic team use RWL [62]. This problem also exists in women and younger age groups [52]. As weight increases, so do the strength parameters in wrestling; wrestlers lose weight because the lower-category wrestlers are physically weaker. Since most wrestlers resort to RWL, those who do not are at a disadvantage because they have to wrestle against much stronger opponents. Since this problem has existed in wrestling for almost 100 years, wrestlers have gotten used to it and do not see it as a problem. The same thing happened in cycling, where doping was so widespread that the cyclists themselves no longer saw it as a problem [51]. Rapid weight gain (RWG) is a phenomenon in which a fighter who has lost weight after the weigh-in gains it back in order to be heavier than their opponent. This is cheating in every sense of the word, as the weight classes were set up so that the athletes fight on a level playing field (not David versus Goliath). When analyzing RWG, we are interested in the amount of mass the wrestler has gained in the time between the weigh-in and the first bout. This amount depends largely on the timing and structure of the competition. In different martial arts, the time varies from 3 to 32 h [52]. A fighter may participate in a league competition and fight only one bout; in an official competition, they usually fight for two days (first day of qualification, second day for rematch and final); appearances in unofficial tournaments can last up to seven days [45]. The problem is that most multi-day events are held with only one weigh-in date before the tournament. This results in a high RWG; at an MMA competition, the average body mass regained was 11.7 ± 4.7% [52]. According to the current rules of wrestling for seniors, a U20 and U23 fighter who weighs 12 kg more should wrestle two or in some cases even three weight classes higher. The World Anti-Doping Agency Code stipulates that a substance or method should be prohibited if it meets two out of the following three criteria: enhances performance, jeopardizes health, or violates the spirit of sport. RWL fulfills all three criteria, and the WADA Code itself suggests that RWL should be banned [51]. Legal regulation on this issue exists only in the United States, specifically for high school wrestlers. It was introduced back in 1989 and was intended to expand throughout the entire United States. Articles that have addressed the results of this regulation are divided, with some claiming that it produces effective results, while others argue that wrestlers evade this regulation. Among the most cited papers on this topic, there is an article that suggests that the same legal regulatory model used in high school wrestling in the United States should be implemented in other combat sports [48]. This model is based on body composition and hydration status. It establishes an individual minimum weight category that is safe for high school wrestlers’ health. This well-designed model has been adopted by other wrestlers, but the question arises as to why it has not been expanded to the rest of the wrestling world from 1989 to the present day (35 years). There were plans to extend it to other wrestling regions throughout the United States by 2005, but these have never been executed. A change in regulations did, however, occur in the **UWW.** Research has shown that in time between the weigh-in and the fights, there is room for manipulation. If there is not enough time, fighters who have lost a lot of body mass cannot recover their performance and do not have time to regain the lost mass (RWG) and do not become heavier than their opponent. Accordingly, an intervention in the rules of the World Wrestling Federation (UWW) happened a few years ago and concerns not only the weigh-in immediately before the competition but adds a weigh-in on every day if the competition lasts several days. “The weigh-in for each category always takes place on the day before the beginning of the competition concerned and lasts 30 min [63]”. “For all competitions, the weigh-in is organized each morning of the concerned weight category. The weigh-in and the medical control lasts 30 min. The second morning of the concerned weight category only the wrestlers who participate in the repechages and finals have to come for the weigh-in [64]“. The rule change shows that the UWW is determined to tackle this problem, but the allowed weight tolerance favors wrestlers who deal with large reductions in body mass. The UWW allowed for a 2 kg tolerance on the second day of competition for non-Olympic events; a 2 kg weight tolerance is allowed for the World Cup, UWW Ranking Series Tournaments, and for International Tournaments [65]. It is worrying that there are many more tournaments that have nothing to do with the Olympic Games and that the occurrence of weight tolerance will be frequent. There is an article on the UWW website that clearly states the position of the UWW Bureau on this issue. The two-kilogram weight tolerance was put forward to the Bureau and it was readily accepted. The move will encourage wrestlers to compete in their preferred weight class instead of an upper weight class in future Ranking Series events [65]. The Bureau members readily accepted the 2 kg tolerance, but the desired category may be 2–3 categories below the realistic one. This approach will not reduce large reductions in body mass. It is interesting that neither WADA nor the UWW Code of Ethics mention RWL and RWG but speak more generally about fair play, prohibited substances, and methods. UWW Code of Ethics 3. 3. 1 states as follows: “The UWW Parties must respect the provisions of the World Anti-Doping Code and of the Olympic Movement Code on the Prevention of the Manipulation of Competitions” [53]. This problem has been discussed for almost 100 years; perhaps it is necessary for the organizations that take care of sports ethics to be more precise on this issue. The question is whether shortening the recovery time (between weigh-ins and fights) is enough to solve the problem of RWL and RWG [50,56]. Some Olympic combat sports other than wrestling have implemented this and no change in habits to reduce body weight has been reported. Future research should re-determine the extent of and trend in body weight reduction to determine whether the rule change had a positive effect on this problem. If the rule change has not been successful, the remaining methods to combat RWL and RWG are to establish a minimum category for each wrestler (legal regulation), eliminate any weight tolerance in all competitions, and more clearly define the problem in the Federation’s code of ethics and the WADA rulebook.

## 5. Limitations of This Study

The study was intended to complement a previous study conducted on a sample of wrestlers, as some highly cited papers on related topics were omitted because they were not conducted on a sample of wrestlers. This study has a problem in that new high-quality papers were not selected due to low citations, and they have low numbers of citations because there was not enough time for a larger number of citations. Another limitation of this study is the small number of papers. According to the draft of this study, we received 60 papers, but a large number of papers were excluded due to the exclusion criteria.

## 6. Recommendations for Healthy Weight Loss

After reviewing studies that focus exclusively on wrestlers, one of the conclusions is that articles with specific recommendations for healthier weight reduction are lacking. The most cited literature on the topic of weight reduction in combat sports provides several pieces of advice on how to safely reduce body weight. Franchini, Artioli, and Raele offer concrete recommendations [44,48,53]. Franchini recommends the following [44]: “Gradual weight loss (i.e., <1 kg.week^−1^), rather than RWL, must be the preferential method for adjusting weight; Athletes should aim to maximize body fat loss and minimize muscle wasting and dehydration when adjusting weight; An athlete who needs to reduce more than 5% of body weight should consider not losing weight: An athlete who needs cut weight so that his/her body fat would lower than 5% for men and 12% for women should consider not losing weight; During the weight loss period, strength training and BCAA supplementation may help preserve muscle mass; Athletes should not undergo low-carbohydrate diets in order to make weight as they seem to be more detrimental to physical performance; If an athlete will have less than 3 h to recovery after the weigh-in, RWL, dehydration and restricted carbohydrate ingestion should be avoided. During the recovery period after weigh-in, athletes are encouraged to consume high amounts ofcarbohydrates, fluids and electrolytes. Creatine supplementation may also be of use if the athlete will recover for a long period after weighing-in. Matches should begin in less than 1 h after weightin; each athlete is allowed to weigh-in only one time; RWL methods and artificial rehydration methods are prohibited on competition days; athletes must pass the hydration test to get the weigh-in validated; an individual minimum competitive weight is determined at the beginning of each season; no athletes are allowed to compete in a weight class that would require weight loss greater than 1.5% of body mass per week”. Artioli recommends the following [48]: “Competition should begin within 1 h after weigh-in, at the latest; each athlete is allowed to be weighed-in only once; rapid weight loss as well as artificial rehydration (i.e., saline infusion) methods are prohibited during the entire competition day; athletes should pass the hydration test to get their weigh-in validated; an individual minimum competitive weight (male athletes competing at no less than 7% and females at no less than 12% of body fat) should be determined at the beginning of each season; athletes are not allowed to compete in any weight class that requires weight reductions greater than 1.5% of body weight per week. In parallel, educational programs should aim at increasing the athletes’, coaches’ and parents’ awareness about the risks of aggressive nutritional strategies as well as healthier waysto properly manage body weight”. Raele [53] has created a decision tree that is relevant to various situations regarding the reduction in body mass. In contrast to Artioli and Franchini, Raele provides more useful information about nutrition under different circumstances of body weight reduction in the paper “Acute-weight-loss strategies for combat sports and applications to Olympic success”.

## 7. Conclusions

From the perspective of motivation, a mental advantage and other issues associated with weight category changes are motives that are not mentioned in studies on wrestlers. Research on wrestlers has reported mild psychological problems with RWL, but the most cited papers suggest that lowering the weight category provides a mental advantage. RWL is harmful to health and should be avoided to the maximum extent possible. Despite numerous warnings, wrestlers continue to use harmful and prohibited methods of weight reduction. Wrestlers who have been reducing significant amounts of body weight over many years will develop various disorders, but this is a long-term problem that wrestlers tend to ignore. Although wrestlers’ adaptation to weight reduction has been mentioned, research has not been able to confirm it. A wrestler does not adapt through experience but rather employs better psychological strategies and weight reduction and recovery strategies. The key to maintaining performance is the time between weigh-ins and the ensuing bouts. If the rules allow more time for recovery, this will open a greater possibility for the manipulation of body weight and the use of harmful methods like infusion. Most of the existing articles agree that it is time for athletes to abandon rapid body weight reductions.

## Figures and Tables

**Figure 1 jfmk-09-00264-f001:**
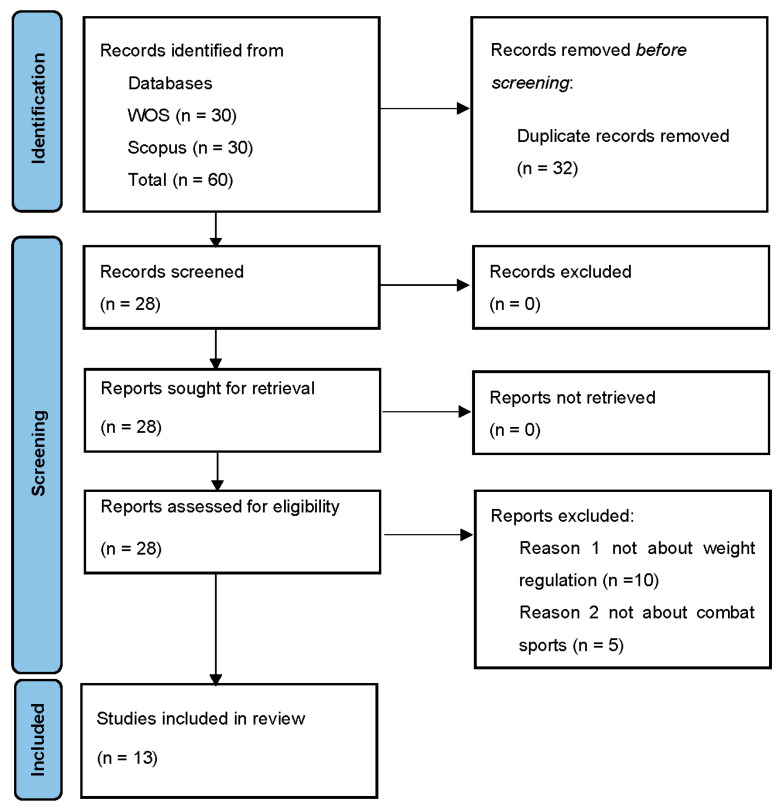
PRISMA flow chart outlining the identification process of the included studies.

**Table 1 jfmk-09-00264-t001:** The most cited articles about weight regulation in combat sport.

	Author	Ref.	Year	Citation WOS	Citation Scopus	Sample	Topic
1	Franchini et al.	[44]	2012	249	260	Combat sports	Weight loss habits, Health
2	Artioli et al.	[45]	2010	195	205	Judo	Weight loss habits
3	Brito et al.	[46]	2012	171	178	Judo, jujitsu, karate and taekwondo	Weight loss habits
4	Artioli et al.	[47]	2010	122	123	Judo	Performance
5	Artioli et al.	[48]	2016	90	99	Combat sports	Legal regulation
6	Pattersson et al.	[49]	2013	89	103	Wrestling, judo, and taekwondo	Psychology
7	Raele et al.	[50]	2017	88	95	Olympic combat sports	Review
8	Artioli et al.	[51]	2010	74	85	Judo	Legal regulation
9	Matthews et al.	[52]	2017	68	46	MMA	Weight loss habits, Health
10	Raele et al.	[53]	2018	68	68	Olympic combat sports	Weight loss habits
11	Mendes et al.	[54]	2013	55	59	Combat sports	Performance
12	Costarelli et al.	[55]	2009	52	62	Judo	Psychology
13	Coswig et al.	[56]	2019	30	38	MMA	Performance

MMA—mixed martial arts.

**Table 2 jfmk-09-00264-t002:** Main findings of 13 selected papers.

	Authors	Title	Main Findings
1	E. Franchini, C. J. Brito and G. G. Artioli [44]	Weight loss in combat sports: physiological, psychological and performance effects	There was a high prevalence (50%) of RWL, regardless of the specific combat discipline. Methods used were harmful to performance and health and included laxatives, diuretics, use of plastic or rubber suits, and sauna. RWL affects physical and cognitive capacities and may increase the risk of death. Recommendations during different training phases and educational and organizational approaches are presented to deal with or to avoid RWL.
2	G. G. Artioli, B. Gualano, E. Franchini, F. B. Scagliusi, M. Takesian, M. Fuchs, and A. H. Lancha [45]	Prevalence, Magnitude, and Methods of Rapid Weight Loss among Judo Competitors	Rapid weight loss is highly prevalent in judo competitors. The level of aggressiveness in weight management behaviors seems to not be influenced by gender or by weight class, but it seems to be influenced by competitive level and by the age at which athletes began cutting weight.
3	C. J. Brito, A. F. C. Martins Roas, I. S. S. Brito, J. C. Bouzas Marins, C. Córdova, and E. Franchini [46]	Methods of Body-Mass Reduction by Combat Sport Athletes	The authors conclude that a high percentage of athletes uses RWL methods. In addition, a high percentage of athletes uses unapproved or prohibited methods such as diuretics, saunas, and plastic clothing. The age at which combat sport athletes reduce BM for the first time is also worrying, especially among strikers.
4	G. G. Artioli, R. T. Iglesias, E. Franchini, B. Gualano, D. B. Kashiwagura, M. Y. Solis, F. B. Benatti, M. Fuchs, and A. H. Lancha [47]	Rapid weight loss followed by recovery time does not affect judo-related performance	Judo-related performance is not affected by an average 5% body weight loss in experienced weight-cyclers if they are able to recover for 4 h. Performance was evaluated through a specific judo exercise, followed by a 5 min judo combat and by three bouts of the Wingate test. No changes were observed in lactate concentration, but a significant decrease in glucose during rest was observed in the weight loss group.
5	G. G. Artioli, B. Saunders, R. T. Iglesias and E. Franchini [48]	It is Time to Ban Rapid Weight Loss from Combat Sports	Athletes use a combination of aggressive and harmful procedures to reduce weight, therefore putting their health at risk, and obtain a competitive advantage against smaller and weaker opponents. Rapid weight loss is unfair because it forces virtually all athletes to reduce weight, causing a cascade effect and risking other athletes’ health. Rapid weight loss meets all three of the World Anti-Doping Agency’s criteria to ban a substance or a method from sport.
6	S. Pettersson, M. P. Ekström and C. M. Berg [49]	Practices of Weight Regulation Among Elite Athletes in Combat Sports: A Matter of Mental Advantage?	Weight regulation has become a key component of the culture of combat sports. Athletes practice weight regulation not only to gain a physical advantage over opponents but also for purposes of identity, mental diversion, and mental advantage. Health care professionals working with weight-category athletes should be familiar with both the negative and perceived positive aspects of weight regulation. Psychological counseling may aid athletes in learning how to gain mental advantages in ways that do not require a focus on weight.
7	R. Reale, G. Slater and L. M Burke [50]	Acute Weight Loss Strategies for Combat Sports and Applications to Olympic Success	While some form of dietary restriction is generally necessary to facilitate AWL (acute weight loss), the most effective strategy to achieve AWL while allowing for the restoration of performance after the weigh-in is to consume strategic amounts of energy from low-weight/low-fiber foods while inducing a mild fluid deficit. Greater fluid deficits and depletion of glycogen stores provide an additional strategy for those requiring greater weight losses. Optimal post-weigh-in recovery strategies are influenced by the method(s) used to achieve AWL.
8	G. G. Artioli, E. Franchini, H. Nicastro, S. Sterkowicz, M. Y Solis and A. H Lancha Junior [51]	The need of a weight management control program in judo: a proposal based on the successful case of wrestling	Weight control program is provided in this manuscript is as follows: competition should begin within 1 h after weigh-in, at the latest; each athlete is allowed to be weighed-in only once; rapid weight loss as well as artificial rehydration (i.e., saline infusion) methods are prohibited during the entire competition day; athletes should pass the hydration test to have their weigh-in validated; an individual minimum competitive weight (male athletes competing at no less than 7% and females at no less than 12% of body fat) should be determined at the beginning of each season; athletes are not allowed to compete in any weight class that requires weight reductions greater than 1.5% of body weight per week.
9	J. J. Matthews and C. Nicholas [52]	Extreme Rapid Weight Loss and Rapid Weight Gain Observed in UK Mixed Martial Arts Athletes Preparing for Competition	The purpose of the investigation was to quantify the magnitude and identify the methods of rapid weight loss (RWL) and rapid weight gain (RWG) in MMA athletes preparing for competition. At the official weigh-in 57% of athletes were dehydrated (1033 ± 19 mOsmol.kg^−1^) and the remaining 43% were severely dehydrated (1267 ± 47 mOsmol.kg^−1^). Results demonstrated RWG greater than RWL. The observed magnitude of RWL and strategies used are comparable to those which have previously resulted in fatalities.
10	R. Reale, G. Slater and L. M Burke [53]	Weight management practices of Australian Olympic combat sport athletes	While many similarities in weight loss practices and experiences exist between combat sports, specific differences were evident. Nuanced, context/culturally specific guidelines should be devised to assist fighters in optimizing performance while minimizing health implications.
11	S. H. Mendes, A. C. Tritto, J. P. L. F. Guilherme, M. Y. Solis, D. E Vieira, E. Franchini, A. H Lancha, G. G Artioli [54]	Effect of rapid weight loss on performance in combat sport male athletes: does adaptation to chronic weight cycling play a role?	Eighteen male combat athletes (WC: n = 10; non-WC: n = 8) reduced up to 5% of their body mass in 5 days. High-intensity intermittent exercise capacity was assessed on a mechanically braked arm ergometer. Chronic weight cycling does not protect athletes from the negative impact of RWL on performance. The time to recover after weigh-in and the patterns of food and fluid ingestion during this period are likely to play a major role in restoring performance to baseline levels.
12	V. Costarelli and D. Stamou [55]	Emotional Intelligence, Body Image and Disordered Eating Attitudes in Combat Sport Athletes	The purpose of this study was to explore the possible differences in body image, emotional intelligence, anxiety levels and disordered eating attitudes in a group of taekwondo (TKD) and judo athletes and non-athletes. Regression analysis revealed that disordered eating attitudes were significantly positively correlated with anxiety levels (*p* < 0.001) and with self-classified weight (*p* < 0.001). Athletes had higher levels of emotional intelligence and a healthier body image compared to non-athletes, but there were no significant differences in terms of disordered eating attitudes.
13	V. S. Coswig, B. Miarka, D. Alvarez Pires, L. Mendes da Silva; C. Bartel, and F. B. Del Vecchio [56]	Weight Regain, But Not Weight Loss, Is Related to Competitive Success in Real-life Mixed Martial Arts Competition	The study aimed to describe the nutritional and behavioral strategies for rapid weight loss (RWL), investigate the effects of RWL and weight regain (WRG) in winners and losers, and verify mood state and technical–tactical/time–motion parameters in mixed martial arts (MMA). RWL and WRG strategies were related to technical–tactical and time–motion patterns, as well as match outcomes. Weight management should be carefully supervised by specialized professionals to reduce health risks and raise competitive performance.

**Table 3 jfmk-09-00264-t003:** Numbers of citations of the top 5 papers for all keyword combinations.

**Web of Science**	11 July 2024						
Keyword 1	Keyword 2	Total results	Paper 1	Paper 2	Paper 3	Paper 4	Paper 5
Weight loss	Combat sport	175	249 *	232	195 *	171 *	152
Rapid weight loss	Combat sport	168	249	195	171	122 *	119
Weight cycling	Combat sport	50	249	152	86	80	72
Weight control	Combat sport	154	232	195	122	100	86
Weight management	Combat sport	65	195	152	79	74 *	68 *
Rapid weight gain	Combat sport	54	90 *	89 *	88 *	68	68 *
**Scopus**	11 July 2024						
Keyword 1	Keyword 2	Total results	Paper 1	Paper 2	Paper 3	Paper 4	Paper 5
Weight loss	Combat sport	180	260	259	178	124	103
Rapid weight loss	Combat sport	119	260	178	124	103	99
Weight cycling	Combat sport	30	260 *	75	68	59	46
Weight control	Combat sport	95	124	113	92 *	90	62
Weight management	Combat sport	38	92	71	68	66 *	38
Rapid weight gain	Combat sport	32	103	99	95	71	59

* Thirteen papers that meet the criteria.

## Data Availability

Data for the current analysis are available upon request and can be obtained by contacting the corresponding author.

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
