# Peer review of "Advice on Regulating Body Mass in Wrestling from the Most Cited Combat Sport Literature—A Systematic Review"

_jfmk, 2024, doi:10.3390/jfmk9040264_

Round 1
Reviewer 1 Report
Comments and Suggestions for Authors
The aim of this paper is to present a review of the most highly cited studies on weight regulation in combat sports. These studies will inform the drawing of conclusions and the formulation of recommendations for wrestlers who manipulate body weight. This paper offers a synthesis of a topic that has been the subject of extensive research and analysis for nearly a century. However, it should be noted that some concerns regarding this work are present:
1. The inclusion criteria defined for this review are not well organized and lack crucial factors needed for this study
2. Abbreviation MMA is used for a type of sport, but the full form for MMA is not described in paper
3. In the results, the table for authors including paper numbers is not organized and even reference numbers are not added to it.
4. The recommendations for regulating the body mass, on which this whole paper is based, not well defined in results and conclusion, lacking the basic theme of writing this paper
5. It is necessary to include a section that sets out the limitations of the study.
Author Response
Rewiever 1
Thank you for the suggestions that certainly improved our paper. All your suggestions have been taken into consideration and added to the paper.
Comment 1
The inclusion criteria defined for this review are not well organized and lack crucial factors needed for this study.
Response 1
The part that talks about the criteria for inclusion in the study has been expanded.
Comment 2
Abbreviation MMA is used for a type of sport, but the full form for MMA is not described in paper
Response 2
The abbreviation is explained in several places
Comment 3
In the results, the table for authors including paper numbers is not organized and even reference numbers are not added to it.
Response 3
Reference numbers have been added to the authors table
Comment 4
The recommendations for regulating the body mass, on which this whole paper is based, not well defined in results and conclusion, lacking the basic theme of writing this paper
Response 4
Instructions for healthy weight loss were in the supplementary materials, but thanks to the suggestion, a chapter on them was added.
Comment 5
It is necessary to include a section that sets out the study limitations.
Response 5
The study limitations section has been expanded
Reviewer 2 Report
Comments and Suggestions for Authors
I ‘ve read with a great interest the article entitled “
Regulating body mass in wrestling, advice from the most cited 2 combat sport literature - Systematic Review. aimed to review the most cited literature on the subject of weight reduction in wrestlers
The introduction provides a good, generalized background of the topic. The aim is clearly defined. The conclusions are consistent with the evidence and arguments presented. Nevertheless the study presents serious limitations and revisions are necessary.
I have the following remarks:
- This review should be characterized as narrative . It would be much more useful if it included quality assessment tools to grade each article.
- Why selecting only the top 5 most cited papers ?I think that is a major limitation of the study and an important source of bias. It is very likely that more recent and more specialized, high-quality articles that simply show different rates of citation have been excluded.
Author Response
Rewiever 2
Thank you for the suggestions that certainly improved our paper. All your suggestions have been taken into consideration and added to the paper.
Comment 1
This review should be characterized as narrative. It would be much more useful if it included quality assessment tools to grade each article.
Response 1
This review was primarily written as a scoping review, but it was suggested to me by the editors that it could be a systematic review. The study also includes review papers, so we gave up on quality assessment, in this case quality is guaranteed by high citations.
Comment 2
Why selecting only the top 5 most cited papers? I think that is a major limitation of the study and an important source of bias. It is very likely that more recent and more specialized, high-quality articles that simply show different rates of citation have been excluded.
Response 2
The basic idea was to obtain 60 papers through the search, it was expected that more papers would remain after the exclusion criteria, but more than half were duplicate papers, at that moment we did not want to give up the draft of the study, I added that to the limitations of the study.
Reviewer 3 Report
Comments and Suggestions for Authors
Review of Regulating body mass in wrestling, advice from the most cited
combat sport literature - Systematic Review
After introduction with very strong aim, I was going forward to read a lot of details about “topic that has been written about for nearly 100 years and investigated in thousands of studies.”
Unfortunately, introduction was only strong part of this article. Here are issues:
- Methodology is not clear. Please write exact algorithm that you put in the search engines – was it “or” between each keyword? I want to replicate your search but your methodology does not allow me.
- Exclusion and inclusion criteria is not clearly stated.
- What did you do if authors does not have concenus? Was decisions blinded as in use in software tools like rayan?
- Why did you omit pubmed?
- Results section needs massive extension and rework. Tables are connected with different type of data and should be separated. It also should be explained and reported with more details.
- It seems like you used discussion for description of results, but still lacking details about procedures.
In conclusion, paper is too general and not insightful. Topic is most welcome in journal but the way that it is presented needs a lot of work in order to be publishable. I suggest authors to dive into details in results and then in discussion try to generalize used strategies and as reference paper possible outcomes for different strategies or findings.
Author Response
Rewiever 3
Thank you for the suggestions that certainly improved our paper. All your suggestions have been taken into consideration and added to the paper.
Comment 1
Methodology is not clear. Please write exact algorithm that you put in the search engines – was it “or” between each keyword? I want to replicate your search but your methodology does not allow me.
Response 1
For example WOS: a combination of two keywords (weight control combat sport) was entered, the search found 157 results, the results are sorted by citations (drop-down menu in Wos), the most cited first. Five studies with the highest citations went into the further process if they fit the criteria. In this case paper 2 and 3 fit the criteria. We have 6 combinations of two keywords, from each we take 5 studies with the most citations. This is now described in more detail in the method chapter.
Comment 2
Exclusion and inclusion criteria is not clearly stated.
Response 2
The criteria for exclusion and inclusion are described in more detail in the methods, I add the table in the supplementary materials, a table with an explanation of the exclusion criteria for each paper in this rewiev.
Comment 3
What did you do if authors does not have concenus? Was decisions blinded as in use in software tools like rayan?
Response 3
Due to the small number of papers and the large number of topics that met the criteria of the study, we did not have such a situation.
Comment 4
Why did you omit pubmed?
Response 4
The sole reason is that Pubmed does not offer the option to sort by citations. Wos and SCOPUS have this type of paper sorting.
Comment 5
Results section needs massive extension and rework. Tables are connected with different type of data and should be separated. It also should be explained and reported with more details.
Response 5
The section has been expanded and the tables are separated.
Comment 6
It seems like you used discussion for description of results, but still lacking details about procedures.
Response 6
The chapter methodes has been expanded, new procedure details have been added.
Round 2
Reviewer 1 Report
Comments and Suggestions for Authors
The critical points have been explained and overcome.
The work can be accepted
Author Response
Thank you for your review.
Authors
Reviewer 2 Report
Comments and Suggestions for Authors
Dear authors. Thank you for your response.There are scales for quality assessment even for systematic reviews.
Nevertheless the manuscript was improved and I have no further suggestions.
Author Response
Thank you for your review.
Authors
Reviewer 3 Report
Comments and Suggestions for Authors
Dear authors,
Check your manuscript again as you forgot to translate some parts from your native language.
I also think that table 2 is not contributing in current form. You did not include all authors and its repetition of some information from table 1. I do not understand your intention. I think you should bo for a column with main findings along each paper or do separate table to show main finding of each paper.
There are few papers. In each section in discussion you could bother to tell i.e what part of performence were affected? You mentioned physiology in this paragraph, but by performance people normally think about sport performance.
But its indeed improved. Work on it once more.
Author Response
Dear reviewer,
Thank you for the suggestions, I hope I have changed everything according to the instructions.
Comment 1
Check your manuscript again as you forgot to translate some parts from your native language.
Respond 1
Thank you for pointing this out, but I have not been able to find these passages even after several readings. I will have the paper proofread before publication. (MDPI English proofreading). I am sure this will solve the problem.
Comment 2
I also think that table 2 is not contributing in current form.
Respond 2
The data from table two has been integrated into the text. Table 2 now contains the key findings you requested in the commentary below.
Comment 3
You did not include all authors and its repetition of some information from table 1. I do not understand your intention. I think you should bo for a column with main findings along each paper or do separate table to show main finding of each paper.
Respond 3
Table 2 now contains the main finding of each paper.
Comment 4
There are few papers. In each section in discussion you could bother to tell i.e what part of performence were affected? You mentioned physiology in this paragraph, but by performance people normally think about sport performance.
Respond 4
This relates to skill degradation. For example, poor results in the Wingate test or the VO2max test. Some papers focus on the relationship between RWL and combat performance, but some also looked at the relationship between RWL and results in different tests.
Round 3
Reviewer 3 Report
Comments and Suggestions for Authors
Indeed now I do not see those parts in native language. As you added new table now I check again and want to clarify some things. In table 3 you mentioned wrong year. Trial is registered in 2024 and not 2014. You should point out search date in methods. Also I had some troubles recreating your results. As you mention you typed keyword. So I go to keyword plus or author keyword section and try to use ("weight loss" and "combat sport"). It results in only 6 matches. But after I typed (weight loss combat sport) in a topic i got similar matches to you, but few months passed and numbers are slighlty different. Please clarify that you search the same way as I did in the attachment or be more straightforward with it. Rest of work is sufficiently improved.

Author Response
Dear Reviewer
The paper has been corrected according to your recommendations, thank you for your effort.
Comment 1
In table 3 you mentioned wrong year. Trial is registered in 2024 and not 2014. You should point out search date in methods.
Response 1
The error in the table has been fixed.
Comment 2
As you mention you typed keyword. So I go to keyword plus or author keyword section and try to use ("weight loss" and "combat sport"). It results in only 6 matches. But after I typed (weight loss combat sport) in a topic i got similar matches to you, but few months passed and numbers are slighlty different. Please clarify that you search the same way as I did in the attachment or be more straightforward with it. Rest of work is sufficiently improved.
Response 2
The answer is in the attached Word document.
